# Soft micromachines with programmable motility and morphology

Hen-Wei Huang[1],*, Mahmut Selman Sakar[2],*, Andrew J. Petruska[1], Salvador Pané[1] & Bradley J. Nelson[1]

Nature provides a wide range of inspiration for building mobile micromachines that can navigate through confined heterogenous environments and perform minimally invasive environmental and biomedical operations. For example, microstructures fabricated in the form of bacterial or eukaryotic flagella can act as artificial microswimmers. Due to limitations in their design and material properties, these simple micromachines lack multifunctionality, effective addressability and manoeuvrability in complex environments. Here we develop an origami-inspired rapid prototyping process for building self-folding, magnetically powered micromachines with complex body plans, reconfigurable shape and controllable motility. Selective reprogramming of the mechanical design and magnetic anisotropy of body parts dynamically modulates the swimming characteristics of the micromachines. We find that tail and body morphologies together determine swimming efficiency and, unlike for rigid swimmers, the choice of magnetic field can subtly change the motility of soft microswimmers.

[1] Institute of Robotics and Intelligent Systems, ETH Zurich, CH-8092 Zurich, Switzerland. [2] Institute of Mechanical Engineering, École Polytechnique Fédérale de Lausanne, CH-1015 Lausanne, Switzerland. * These authors contributed equally to this work. Correspondence and requests for materials should be addressed to B.J.N. (email: bnelson@ethz.ch).

Unicellular microorganisms exhibit several different flagellar mechanisms that couple with various distinct body forms and structural organizations to enhance motility[1] (Fig. 1a). Many of these microorganisms are able to drastically change their morphologies and motion mechanisms to adapt to changes in their microenvironment. For example, *Trypanosoma brucei*, the cause of African trypanosomiasis or sleeping sickness, exhibit a long slender form with a free flagellum that propels the organism through bodily fluids and penetrates the blood vessel endothelium to invade extravascular tissue[2]. When the bacterium enters the blood stream, a quorum sensing-like mechanism transforms the long slender form into a shorter stumpy morphology to aid in its survival in locations where providing its own motility is less important (Fig. 1b). The microorganism is capable of transforming itself back and forth between forms depending on its local environment.

While flagellated cells have been the primary source of inspiration for building artificial microswimmers[3,4], the construction of bioinspired motile micromachines has focused solely on the flagellar propulsion mechanism and has ignored the coupled effects between the cell body and the flagella. Magnetic helical microswimmers, for example, have been engineered by artificially approximating the flagellum of *Escherichia coli*[5]. These rigid helical structures propell through the medium in a corkscrew like motion and can be manufactured with a number of techniques such as self-scrolling[6], template-assisted electrodeposition[7], glancing angle deposition[8] and direct laser writing[9]. Alternatively, following the example of eukaryotic flagella, microscopic filaments with flexible joints have been fabricated[3,10], which can be propelled by inducing a beat pattern with an oscillating magnetic field. These prototypes experimentally realized controlled swimming motion at low Reynolds number, but motile cells are capable of more than swimming, as in the case of *T. brucei*. Microorganisms like these can adapt their shape and respond to their environment by exploiting the soft and stiff structures within their complex body plans.

In order to expand the potential of biomimicry in the domain of synthetic microorganisms, we introduce a manufacturing process for building reconfigurable motile micromachines with compound bodies that respond to external control signals. The integrated manipulation platform provides automated magnetic control for mobility and spatiotemporally controlled heating for shape shifting. We demonstrate that, through transformation of morphological state, the characteristics of mobility can also be modulated. In accordance with the success of soft robotics at macroscale[11–13], our results show that autonomously assembled soft micromachines composed of smart materials have the potential to exhibit superior functionality and adaptation.

## Results

### Design and manufacturing of soft compound micromachines.

We use lithographic patterning of hydrogel sheets as a rapid and high-throughput method for creating compound micromachines from biocompatible materials. Hydrogels can form complex three-dimensional (3D) structures through stress-induced bending[14,15]. At the microscale, these deformations can be programmed through the construction of layered materials[16], the formation of interlayer bands with different material properties[17] or the generation of internal monomer gradients[18]. At the mesoscale, similar deformation protocols can be generated by patterning stiff reinforcing elements within a soft matrix[19]. In this work, we combine multilayered patterning and local reinforcement to precisely and independently control the folding behaviour for each component (Fig. 1c). The structures consist of a non-swelling supporting hydrogel layer composed of poly (ethylene glycol) diacrylate (PEGDA) selectively patterned on a swelling thermo-responsive *N*-isopropylacrylamide (NIPAAm) hydrogel layer. The motility of these devices is induced through the addition of magnetic nanoparticles (MNPs) in the hydrogel layers and tailored by magnetically aligning the particles during polymerization[20]. Selective alignment of MNPs in different compartments results in a multitude of magnetic axis. MNPs are also utilized as reinforcing components, and the folding axis of each compartment can be programmed through selective alignment of MNPs. Consequently, self-folding transforms a two-dimensional hydrogel microstructure into a 3D biomimetic microswimmer with tailored shape and magnetic anisotropy (Fig. 1d).

Since the micromachines are sculptured from stimuli–responsive hydrogels[21], multiple morphologies and motilities can be programmed for a single device. In our previous work, we achieved near-infrared (NIR) responsive devices made of a nanocomposite consisting of an hydrogel matrix containing graphene oxide nanoparticles[22]. In this work, we replace the graphene oxide nanoparticles with MNPs, which provide similarly enhanced NIR heating[23]. The spontaneous folding and temperature-dependent morphologies of the preprogrammed monolayer and bilayer sheets create multifunctional compound micromachines with controllable structural characteristics without the need for further assembly or reassembly steps (Fig. 1f–h). The motility strategies of these micromachines can be modulated through external stimuli that excite different interactions with the environment or affect the swelling characteristics of the hydrogel and, in turn, the overall morphology of the device (Fig. 1e).

### Programming machine architecture through particle alignment.

While photolithography provides an efficient method to pattern the overall blueprint of the compound micromachine, tuned reinforcement of a soft nanocomposite matrix provides a method to control the bending direction of individual building blocks. Rectangular bilayer sheets fold along their long side in the absence of particle alignment due to the edge effect[24]. The edge effect is highly dependent on the sheet's boundary conditions, which limits the design of compound machines that have multiple interconnected parts. To demonstrate the effect, a flagella structure was fabricated at different locations along a rectangular plate (Fig. 2a). When the flagellum was positioned in the middle of the long edge, the plate folded into a tube about its short edge. In contrast, placing the flagellum at the corner caused the plate to fold into a helix. The subsequent analysis of stress distributions, calculated using finite element modelling, corroborated the observed bending modes (Supplementary Note 1). Thus, to create a reliable folding protocol that is independent of the flagellum location, an additional folding 'programming technique' is required. This can be achieved by incorporating MNPs within the supporting hydrogel nanocomposite that are aligned by an external magnetic field (Supplementary Fig. 1). The orientation of the alignment sets the anisotropic expansion of the bilayer and, therefore, the folding axis[19]. In these structures, the folding axis is perpendicular to the alignment of the MNPs in the supporting layer (Fig. 2b,c). Using a planar alignment of particles in the supporting layer, elongated bilayer tubes with polar flagella attached to their distal end were fabricated (Fig. 2b). Likewise, the folding axis of monolayer structures can be determined by controlling MNP alignment (Fig. 2d). By tuning the alignment of particles in both the head and the flagellum, we fabricated a variety of bioinspired flagellated micromachines (Supplementary Fig. 3).

**Programming triggered transformation of morphology**. Since these devices are composed of thermally responsive hydrogels, the folded shape is sensitive to changes in ambient temperature. NIR laser heating is a convenient mechanism for stimulating these morphological changes, because adjusting the exposure area provides spatiotemporal control over stimulation and selectivity at the component level[23]. Unfortunately, the hydrogel is mostly transparent to these wavelengths, thus the MNPs must convert the NIR energy into heat. The NIR heating performance of the MNP-hydrogel composite structures was verified using a 808 nm

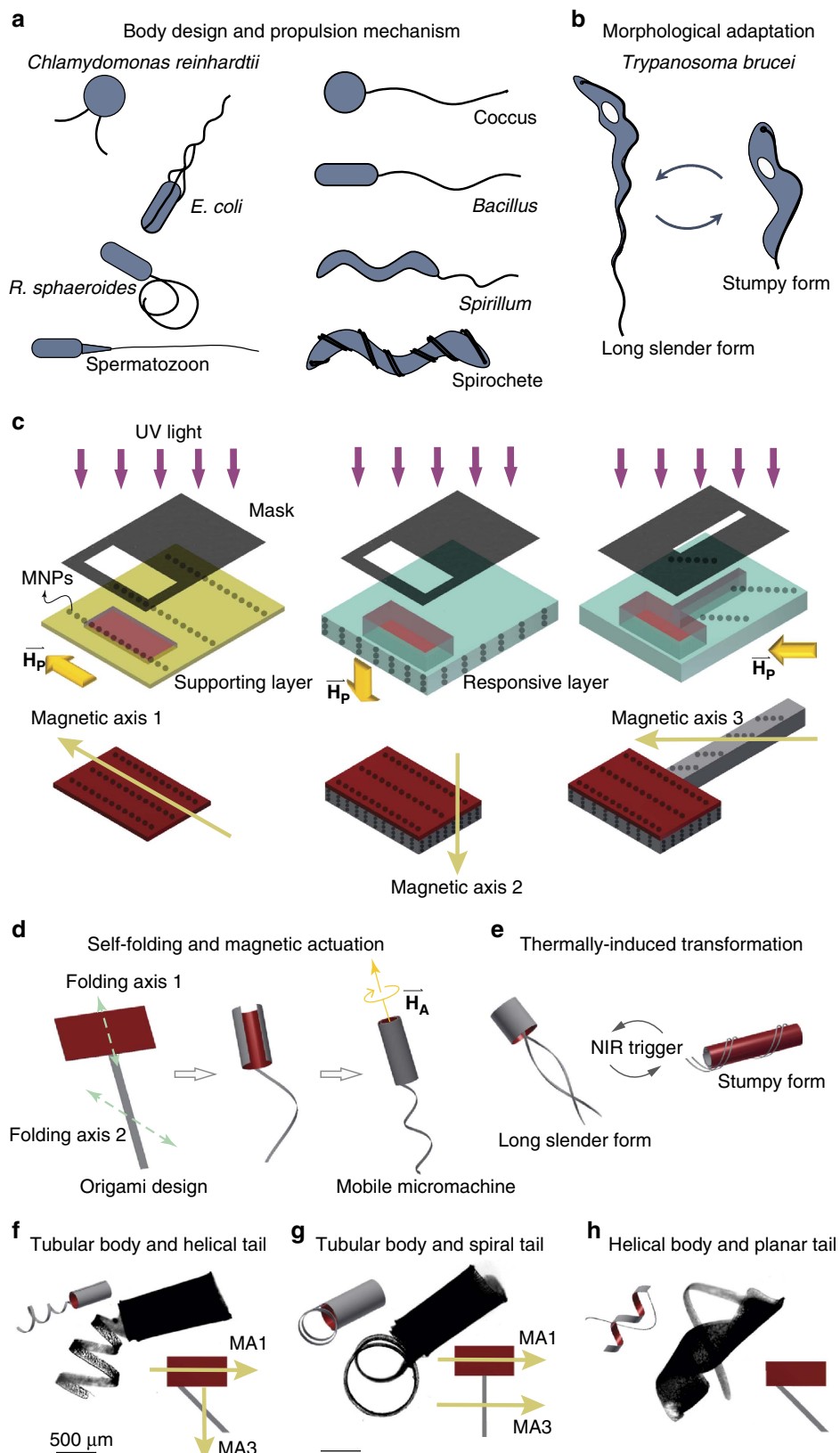

wavelength source. It induced a 25 °C temperature rise in the nanocomposite within 15 s (Supplementary Fig. 4a). In bilayer structures, the differential swelling response of individual layers completely unfold the structure into a flat sheet at 40 °C. This structure begins to refold, in the opposite direction and about a perpendicular folding axis, above 42 °C and forms a different tubular shape by 45 °C (Fig. 3a, Supplementary Movie 1 and Supplementary Fig. 4).

The shape transformation is completely reversible and the micromachines will eventually unfold and fold back to their initial state when they are cooled down. The transition temperature can be adjusted by tuning the type (hydrophobic or hydrophilic) and concentration of co-polymers incorporated into the nanocomposite[25] (Supplementary Note 2). For example, by removing the hydrophilic co-polymer (Acrylamide, AAm) from the nanocomposite solution, we can decrease the transition temperature from 45 to 32 °C (Supplementary Fig. 5). Using NIR, a machine can be selectively heated up to completely transform its shape from one state to another in 65 s. Immediately after the halt of exposure, the machine cools down and transforms back to its original shape in 74 s (Supplementary Fig. 4f). The temperature-dependent radii of bilayered structures were analytically predicted using modified Timoshenko bimorph beam theory[26]:

$$R = \frac{(h_1 + h_2)\left(8(1+m)^2 + (1+mn)\left(m^2 + \frac{1}{mn}\right)\right)}{6\varepsilon(1+m)^2} \quad (1)$$

where $n = E_1/E_2$ is the ratio of elastic modulus of the non-swelling supporting layer to swelling layer, $m = h_1/h_2$ is the thickness ratio of the two coupled layers, and $\varepsilon$ is the difference in expansion coefficient between the two coupled layers. The estimated values of the radius closely agree with the measurements for different temperature values during unfolding (Fig. 3b). In monolayer structures, the heating affects the volume of the feature but not its form. The temperature-dependent radii of monolayer structures are given by $R = R_0(1 + \alpha_T \cdot \sin\theta \cdot \Delta T)$, where $\alpha_T$ is the negative thermal expansion coefficient of the nanocomposite, $\Delta T$ is the temperature change, and $R_0$ is the radius at room temperature.

**Programming magnetic anisotropy in self-folding micromachines.** The MNPs not only encode the folding characteristics and allow for NIR heating but also enable the motility of the hydrogel structures. MNP distribution in the hydrogel matrices is crucial for establishing the alignment of the device with the applied magnetic field. When the MNPs are randomly distributed, the micromachine behaves like a metalic rod and aligns with its long axis along the magnetic field. Such a device will tumble and not achieve a corkscrew motion when exposed to a rotating magnetic field. To achieve corkscrew motion, the radial direction of the micromachine needs to be aligned with the field once folded. Although a planar alignment of MNPs can result in the micromachine folding along its long axis and magnetically aligning perpendicular to this axis (Supplementary Fig. 6), the alignment becomes parallel with the long axis of the device once heated and refolded (Fig. 3c). In their refolded states, these machines switch to a tumbling motion from the programmed corkscrew motion. Setting out-of-plane alignment of the MNPs in the swelling layer does not affect the folding characteristics but does ensure a radial magnetic alignment once folded or refolded (Fig. 3c), which enables corkscrew propulsion. Since the magnetic moment of the machine is a function of the MNP volume[27], the resulting motion is dominated by the component with the most MNPs. Thus, the head contributes more torque than the flagella, and within a bilayer the swelling layer contributes more than the supporting layer due to higher nanoparticle concentration. The magnetic moments of nanocomposites with different MNP volumes are shown in Supplementary Fig. 6c. As a result, temperature-dependent morphology can be programmed independently from magnetic alignment, which can be specified separately for the flagella, the supporting layer and the swelling layer.

**Propulsion of soft compound micromachines.** Since the magnetization and morphology of each component can now be specified separately, we were able to fabricate machines with identical bilayer tubular heads, but with either planar or helical flagella in order to explore the role of flagellar morphology on propulsion. The dimensions of the fabricated hybrid machines are given in Supplementary Fig. 3a and Supplementary Table 1. The folded machines were suspended in a viscous sucrose solution (4 cP) that provided an environment with a Reynolds number between $10^{-1}$ and $10^{-3}$. The magnetic fields were generated using an eight-coil electromagnetic manipulation system that is capable of independently controlling the force and torque acting on the machine to generate propulsion[28]. A rotating magnetic field that has no gradient and, therefore, applies no forces is used for the propulsion experiments in this study. Previous analysis showed that this mode of actuation is more versatile and efficient for *in vivo* applications[29]. A time-lapse of the trajectories is shown in Fig. 4a. The 3D helical trajectories of micromachines were tracked using two orthogonal cameras and identified two initially unexpected behaviours.

The first behaviour that we observed was that the motility of the micromachine with the flat flagellum was significantly

**Figure 1 | Reconfigurable body plans for soft micromachines inspired from microorganisms.** The versatile body plans of microorganisms composed of (**a**) a variety of body design and propeller mechanisms and (**b**) materials and architectures that can be utilized for morphological adaptation. (**c**) Schematic of the batch fabrication of biomimetic soft micromachines. This process enables photopatterning of microstructures with various shapes, as well as the ability to fix the magnetic nanoparticles (MNPs) in the structure. Flagellated soft micromachines with a bilayer head and monolayer tail are fabricated by sequential photopatterning of magnetic hydrogel nanocomposites. First, a mixture of photocurable non-swelling hydrogel and MNPs is injected into a microfabricated chamber (constituting the supporting layer) and a uniform magnetic field ($\overrightarrow{\mathbf{H_P}}$) is applied in direction 1. Second, a mixture of photocurable swelling thermo-responsive hydrogel layer is patterned on top of the supporting layer and a uniform magnetic field is applied in direction 2. Finally, a monolayer tail embedded with aligned nanoparticles in direction 3 is attached to the previous bilayer structure using the same process. Every layer has its own fixed magnetic axis denoted by magnetic axis 1 (MA1), magnetic axis 2 (MA2) and magnetic axis 3 (MA3). (**d**) Anisotropic swelling behaviour controlled by the alignment of MNPs along prescribed 3D pathways and selective patterning of supporting layers results in 3D functional micromachines. The folding axis 1 and folding axis 2 denote the direction of folding for each compartment. The micromachine possesses multiple different magnetic axes, which determine the motility when the magnetic field is applied. The flagellated micromachine, which contains self-assembled MNPs, performs controllable swimming in a 3D space under a homogeneous rotating magnetic field ($\overrightarrow{\mathbf{H_A}}$). (**e**) The soft micromachine can be programmed to transform its shape and perform a different propulsion mechanism when exposed to external near-infrared (NIR) heating. (**f-h**) Optical images of flagellated soft micromachines with complex body plans. MA1 and MA3 denote the magnetic axis in the head and the tail, respectively. Scale bars, 500 μm.

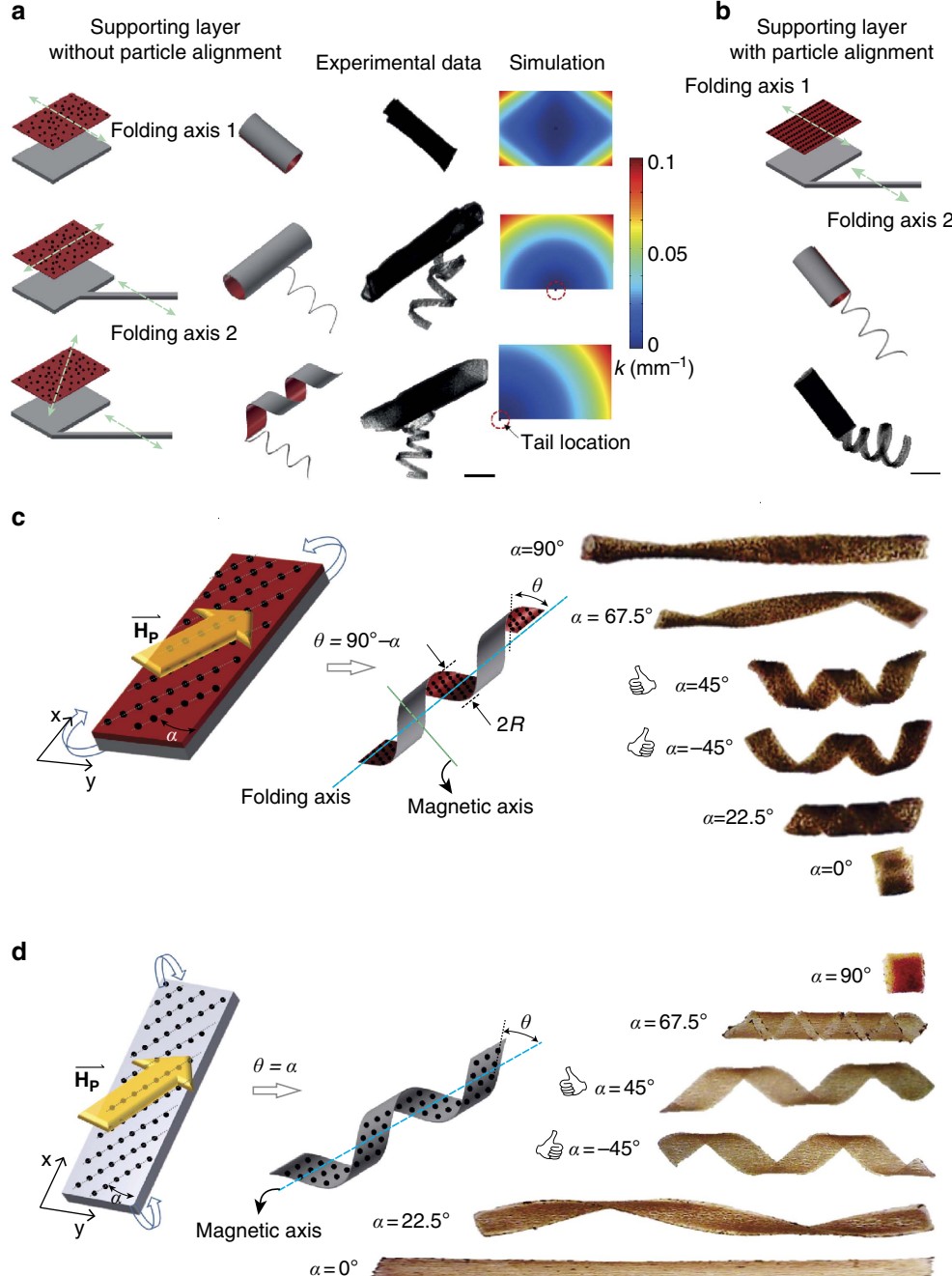

**Figure 2 | Programming the morphology of the flagelled soft micromachines.** (**a**) Formation of compound micromachines with a bilayer head and monolayer tail. Due to lack of particle alignment, the folding axis of the head is solely determined by the edge effect. Finite element modelling (FEM) simulations visualize internal stress distribution. (**b**) By reinforcing the magnetic nanoparticles (MNPs) inside the supporting layer, the folding of the head can be decoupled from the geometric effects and directly controlled by the particle alignment. Scale bars, 500 μm. The final 3D shape of the helical structures after autonomous folding of hydrogel (**c**) bilayers and (**d**) monolayers is controlled by the alignment of embedded MNPs. The orientation of MNPs alignment (denoted by α) generates a helical angle (denoted by θ) given by $\theta = 90° - \alpha$ in bilayer structures and $\theta = \alpha$ in monolayer structures. Scale bars, 1 mm.

different than the micromachine with the helical flagellum. While chirality of the helical flagellum generates a propulsive force by breaking the time-reversal symmetry, the planar flagellum acts like a flexible oar that deforms the whole body and results in forward motion. The frequency versus forward speed along the axis of helical trajectory of different prototypes is given in Fig. 4b. The maximum dimensionless speed ($U/fL \times 10^3$) of the micromachines with planar flagella was observed to be 375 and occured at 2 Hz (Supplementary Fig. 7a), where $U$ is the forward velocity,

$f$ is the rotating frequency and $L$ the length of machine head. This is three times faster than the devices fabricated with helical flagellum and significantly faster than other wirelessly powered micromachines with flexible propellers reported in the literature[30].

Theoretical analysis and experiments with flagellated bacteria have shown that the propulsion of a helix in a viscous fluid produced by its rotation is very inefficient, both kinematically (small forward motion per rotation) and energetically (only 2%

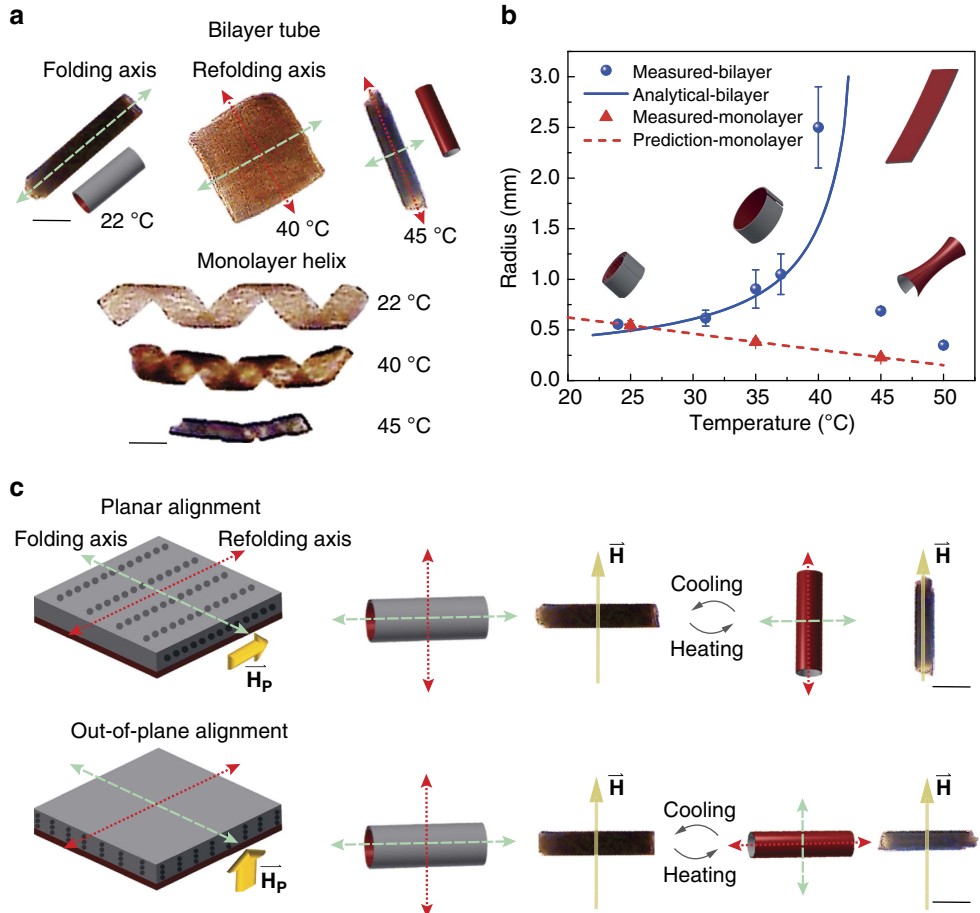

**Figure 3 | Programmable transformation of morphology and magnetic anisotropy.** (**a**) Optical images of the bilayer tube and monolayer helix (with a helical angle of 45°) at different ambient temperatures. (**b**) The measured and predicted folding radius of the monolayer and bilayer structures at different temperatures. Each dot represents the average radius for three different machines with identical cylindrical shapes ± s.e.m. (standard error of the mean). (**c**) Programming the magnetization of micromachines generated after shape transformation. The alignment of the magnetic nanoparticles (MNPs) embedded inside the responsive layer determines the magnetic axis of soft micromachines in their transformed state. While magnetization generated with planar alignment of MNPs can be reconfigured during the refolding process, the machines with magnetization in the out-of-plane direction display the same magnetic axis. A static uniform magnetic field ($\vec{\mathbf{H}}$) is applied to identify the magnetic anisotropy. Scale bars, 1 mm.

percent of the total work done by the helix is transmitted to propulsive work)[1]. Experiments with uniflagellated bacterium *Caulobacter crescentus*[31] supported the idea that the flexibility of the hook may contribute to efficieny of flagellar swimming. Due to the bending of the hook, the cell body is tilted with respect to the direction of motion, and it precesses, tracing out a helical trajectory. The helical motion of the cell body generates thrust and enhances forward swimming motility, which is proportional to the precession angle. We did not engineer a special hook that connects the tail to the head. For the chosen material composition and design parameters, micromachines with helical tails exhibit high symmetry in rotating axis, which stabilizes the precession of head and results in a relatively straight trajectory (see Supplementary Movie 2). Since the precession angle is proportional to the radius of helical trajectory, the angle determines how the dynamic deformation of the body transforms into forward motion, a larger precession angle equates to faster propulsion. Thus, the superior swimming performance of micromachine with planar tail is induced by a coupling affect between the flexible flagellum and tubular head that increases the precession angle of the motion. Increasing the frequency of the rotating magnetic field will diminish the precession angle of the machine and decrease the forward

velocity, which is also consistent with our observations (Fig. 4b, Supplementary Fig. 7b and Supplementary Movie 3). To further verify that propulsion is due to the coupling between the flagellum and the body and not a magnetic affect, the flagella were removed from the compound micromachines. Without the compound geometry, the devices rolled in place and did not propell forward (Supplementary Movie 3).

The second unexpected behaviour was that the swimming behaviour of the flagellated micromachines depends on both the rate and direction of the magnetic-field rotation. The 3D trajectories showed that both devices followed helical trajectories with the head of the machines aligned tangent to this trajectory (Fig. 4c,d). Initially, the chirality of the flagellum matched the chirality of this helical trajectory, so both the head's and the flagellum's motion contributed to forward propulsion (Supplementary Movie 2). Unlike other flagellum based micromachines[6], which are rigid and do not deform during locomotion, when the rotational direction was reversed, the swimming direction of the soft micromachines remained unchanged. The chirality of the resulting trajectory did, however, reverse. While the machine is moving forward, the body is tilted with respect to the direction of motion. In the reverse mode, the precession still points in the forward direction.

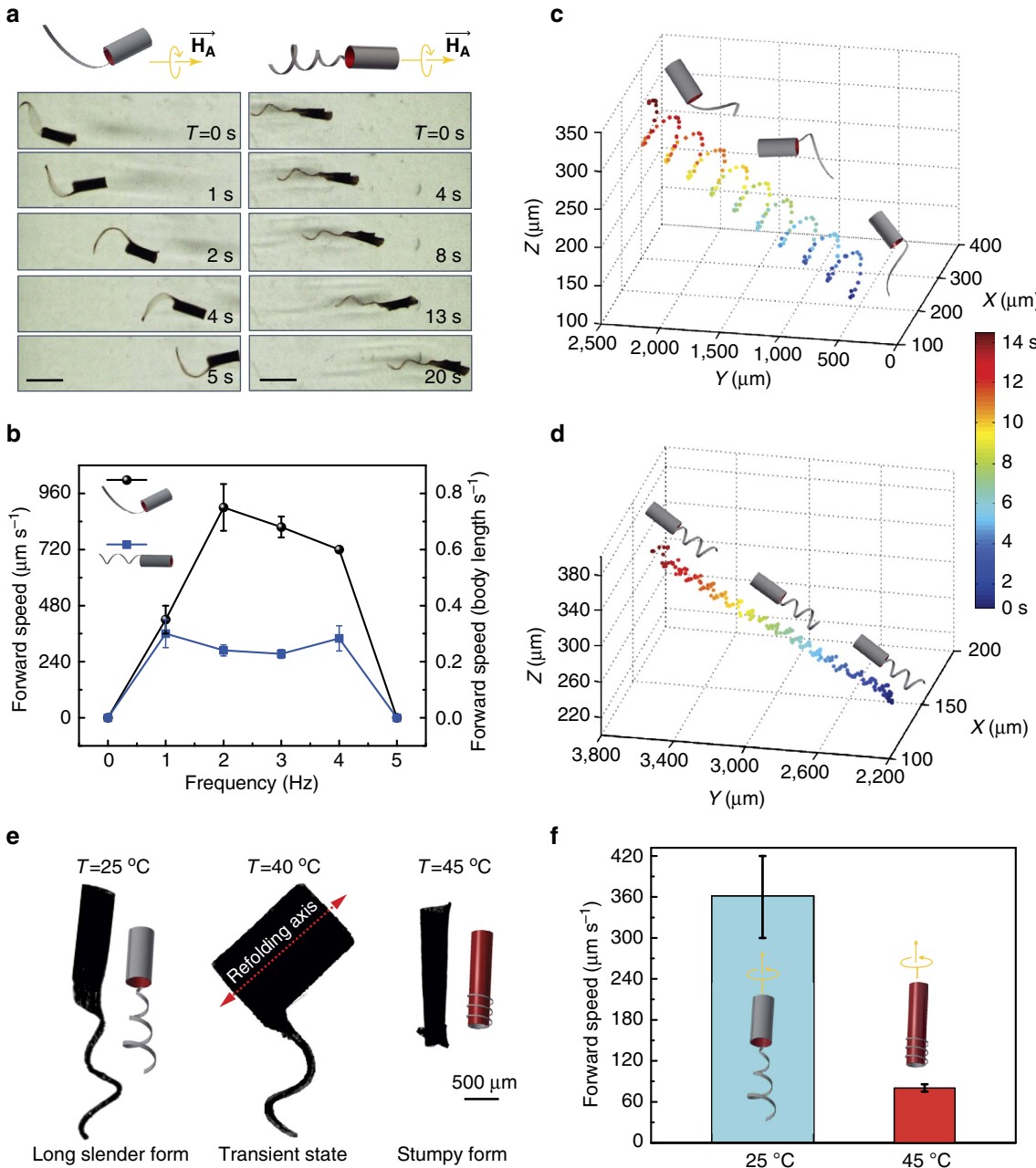

**Figure 4 | Programming the motility of flagellated soft micromachines. (a)** Time lapse optical images of two different types of compound micromachines driven by rotating uniform magnetic fields. Scale bars, 2 mm. **(b)** The forward velocity of the compound machines for micromachines with planar and helical flagella with respect to the frequency of rotating magnetic field. The 3D trajectories of the flagellated soft machines with **(c)** planar flagellum and **(d)** helical flagellum recorded by two camearas visualizing the workspace from the top and side view. **(e)** A reconfigurable compound soft micromachine switching shapes between a long slender form and a compact stumpy form. **(f)** The forward speed of the micromachine at different morphological states driven with the same strength and frequency of rotating magnetic field. All bar graphs represent average ± s.e.m. ($N = 6$ measurements per micromachine, three different micromachines).

As the helical flagellum and the trajectory now have opposite chiralities, the flagellum's motion retards the machine's progress. This implies that, similar to the machine with an oar-like flagellum, the precession driven by the coupling between the head and flagellum dominates the propulsion at low rotating frequency. Due to the attenuation of motion in the reverse mode, the machine starts to move backwards only when the corkscrew motion is stronger than the precession, which only happened at increasing frequency. Likewise, increasing the rotational frequency reduces this affect and causes the

micromachine to change swimming directions as the flagellum's propulsive force dominates (Supplementary Fig. 7c and Supplementary Movies 4 and 5).

To systematically demonstrate the contribution of the precession, we fabricated three different micromachines: a non-deformable (relatively stiff) machine with only a helical body, a non-deformable machine with helical tail and tubular head, and a deformable (relatively soft) machine with helical tail and tubular head (Supplementary Fig. 8 and Supplementary Movie 5). The machines were driven at 2 and 3 Hz in clockwise (CW) and

counterclockwise (CCW) direction. The first and second type of micromachines instaneously changed direction of motion when the direction of magnetic field rotation was reversed even at very low frequencies. The existence of a tubular head generates higher drag force, which reduces the motility in both directions. Therefore, micromachines with tubular heads swim with lower velocity than the machine with helical body. The motility of these soft micromachines is comparable to rigid micromachines. The motility of the third type of soft micromachine is composed of the precession of the body and the procession generated by the tail. These machines follow a helical trajectory, and the precession enhances the forward motility (CCW rotation) while attenuating the backward motility (CW rotation) as expected. At low rotating frequencies, the motility of the machine is dominated by the precession of the body because of the large precession angle and low propulsion force generated by the tail. As a result, the machine moves forward regardless of the direction of magnetic field rotation (CW and CCW). Increasing the frequency reduces the precession angle and increases the propulsion forces of the tail. Therefore, the motility is dominated by corkscrew motion at high frequencies and this motion is reversible (machines reverse their direction when the direction of rotation is reversed). Thus, these compound micromachines have multiple motility strategies that can be tuned through design and selected through applied magnetic fields.

**Programming transformation between propulsion strategies.** While the choice of magnetic field rotation rate and frequency can subtly change the device's morphology and motility, NIR heating induces fundamental changes in the micromachine's structure. This can be leveraged, for example, to wrap the body around the flagella in order to hide it from the environment or even to release backup flagella if the primary is removed or damaged (Supplementary Fig. 9). Although such complex designs proved to be highly senstive to small variations in manufacturing steps and difficult to reliably produce, more robust transformations such as wrapping the flagella about the cylindrical head are possible. We engineered hybrid machines composed of bilayer tubular heads and monolayer helical flagella. The machine head was encoded with a radial magnetization by aligning MNPs in the out-of-plane direction. When driven by rotating magnetic fields, the machines rotate around their long axis and perform corkscrew motion. The micromachine can then reversibly switch its morphology between a long slender form and a stumpy form when exposed to NIR heating (Fig. 4e). When the whole body was exposed to NIR light, the tubular head unfolded and the helical flagellum shrank. Further heating initiated the refolding of the head, and due to the location of the flagellum, the fluidic forces during rotation of the refolded machine wrapped the flagella around the tip of the head. The long slender form and the short stumpy form had a significantly different forward speed (Fig. 4f). While the long slender form was engineered to swim in free space, the short stumpy form allows the machine to hide its flagellum from adhering to the walls of narrow cavities or densely packed suspensions. The reconfiguration of shape can also be utilized to reprogram the magnetization of the machine (as shown in Fig. 3c) in such a way that it can tumble on planar surfaces in the stumpy form and swim in the elongated form (Supplementary Movie 6).

## Discussion

In nature, motile organisms are able to drastically change their morphologies and motion mechanisms to adapt to changes in their microenvironment. *African trypanosomes* exhibit a long slender form with a free flagellum that drives the organism through bodily fluids and penetrate the blood vessel endothelium to invade extravascular tissues[2]. Upon entering the blood stream, a quorum sensing like mechanism transforms the long slender form into a shorter stumpy form for survival in the tsetse fly (Fig. 1b). When an infected host is bitten by a tsetse fly, parasites are taken up with the blood meal, where short stumpy forms differentiate into a highly migratory phenotype and establish a midgut infection. Building micromachines that can change the morphology and architecture of their bodily parts will create opportunities for engineering plasticity and adaptation in synthetic systems. The shape of the machines governs the response to the magnetic input due to hydrodynamic forces. As a result, the motility of shape shifting micromachines can be adapted to different complex environments. In addition to biomimetic strategies, novel transformable modules can be invented to enhance motility in certain target locations. For instance, transforming a conical head into a helical one may enable superior mobility in highly viscous fluids (as mastered by *Spirochetes*[32], see Fig. 1a), while transforming back to the original shape may facilitate tissue penetration. Reconfigurable soft compound micromachines can realize these complex biological features.

In summary, we present a practial method for building functional micromachines with complex geometries through programmable self-assembly. Controlling the morphology of the folded mechanism by patterning multiple hydrogel layers limits the scalibility of the approach, as this technique requires patterning of high aspect ratio structures. Generation of stiffness gradients within a hydrogel monolayer with gravitational sedimentation allow us to initate self-folding[33], and more sophosticated devices at the microscale can be fabricated by magnetically controlling the distribution of MNPs along the thickness direction. Although we focus on bioinspired corkscrew motion, our approach is not limited to propulsion generated by a rotating passive flagellum. More complex swimming strategies can be realized by engineering parts with a different magnetic easy axis[20] and application of spatiotemporally varying magnetic fields. NIR exposure introduces high precision addressing of individual micromachines and the use of shape as an additional degree of freedom to control propulsion. While thermally triggered deformation is completely reversible for the presented prototypes, micromachines with multiple stable conformations can be engineered by combining local heating with engagement initiated by internal magnetic forces among body parts[34]. The NIR laser spot can be adjusted to be small enough to ensure selective exposure of components as shown in Supplementary Fig. 4d and Supplementary Movie 6.

## Methods

**Materials.** *N*-Isopropylacrylamide monomer (NIPAAm), AAm, Poly (Ethyle-neglycol) Diacrylate (average MW 575, PEGDA) and 2, 2-dimethoxy-2 phenyla-cetophenone (99%, DMPA), ethyl lactate (98%, EL) were obtained from Sigma-Aldrich and used as received. SU-8 photoresist was purchased from Microchem (USA), while S1813 photoresist and MF319 developer were purchased from Clariant (Germany). Dispersible 1% polyvinylpyrrolidone (PVP) coated 30 nm magnetite ($Fe_3O_4$) was purchased from Nanostructured and Amorphous (USA).

**Experimental platform.** The 3D alignment of the MNPs was performed using a solenoid with an inner diameter of 5.5 inch in conjunction with a pair of Helmholtz coils separated by 5.5 inch. Ultraviolet lamps (Lightning Enterprises, USA) were integrated inside the solenoid to initiate the cross-linking of hydrogel polymer. Images of the magnetic setup are shown in Supplementary Fig. 1. The maximum strength of generated uniform magnetic fields by the solenoid and Helmholtz coils are 10 and 6 mT at the center region, respectively. The motion studies were conducted with an eight-coil electromagnetic manipulation system called the Octo-mag[28]. The maximum uniform magnetic field generated by the system is 40 mT, and the maximum magnetic field gradient is 1 T/m.

**Formulation and preparation of pre-gel solutions.** Responsive gels were fabricated using ultraviolet-assisted polymerization. N-isopropylacrylamide (NIPAAm) was used as the monomer, while AAm and PEGDA were used as the hydrophilic co-monomer and the cross-linker, respectively. To obtain a strong temperature induced swelling response and a lower critical solution temperature of 42 °C, which is slightly higher than the physiological temperature of the human body, the molar ratio of NIPAAm–AAm–PEDGA was adjusted to 85:15:1 (Supplementary Note 3 and Supplementary Fig. 5). For each 10 g of NIPAAm–AAm–PEDGA 0.3 g of 2, 2-dimethoxy-2 phenylacetophenone (DMPA, photoinitiator) and 7 g of EL (solvent). The mixture was subject to an ultrasonicator bath for 20 min until complete dissolution. Subsequently, 0.865 g of MNPs, $Fe_3O_4$ with 30 nm diameter, were added per each 17.3 g of the pre-polymer solution and dispersed by probe ultrasonication at 8000J (SONICS, USA). The non-responsive gel PEGDA was incorporated as a supporting layer for bilayer structures. Per each 10 g of PEGDA, 0.3 and 5 g of DMPA and EL were added, respectively, and dissolved by means of bath sonication for 10 min. To 15.3 g of this solution, 0.19 g of MNPs were added into the passive gel and dispersed by probe ultrasonication at 2000 J. The details of the characterizations of hydrogel nanocomposites are provided in Supplementary Note 2 and Supplementary Fig. 10.

**Patterning hydrogel nanocomposite layers using photolithography.** A plastic photomask foil was designed and printed (Selba SA, Switzerland). The designed features were transferred onto a glass wafer by patterning S1813 photoresist via lithography. Subsequently, a 100 nm thick layer of Cr was deposited on the glass wafer, and a lift-off process was performed to obtain the final glass photomask. Patterning SU-8 photoresist on silicon substrates produced spacers of defined thickness. Evaporation of silane groups generates a non-adhesive surface. The schematics of the sandwich configuration are shown in Supplementary Fig. 1b. The substrates are reusable after cleaning with acetone, isopropyl alcohol (IPA) and deionized (DI) water.

**Fabrication of monolayer structures.** For the fabrication of monolayer machines, a pre-gel nanocomposite solution (NIPAAm–AAm–PEDGA and MNPs) was injected into the 30 µm thick chamber constructed by sandwiching a glass mask and a silicon wafer with patterned SU-8 spacers. After the pre-gel solution filled the entire chamber, it took ∼1 hour to generate a gradient of MNPs through gravitational sedimentation. Subsequently, a static uniform magnetic field with an intensity of 10 mT and ultraviolet exposure (365 nm, 3 mW cm$^{-2}$) for 1 min were simultaneously applied to the pre-gel solution in order to align the MNPs while polymerizing the pre-gel solution. After ultraviolet curing, the sandwich construction was opened. By adding water, the structures were released from the glass mask due to swelling.

**Fabrication of bilayer structures.** For the fabrication of bilayer machines, first a supporting layer was obtained by placing a passive pre-gel solution (PEGDA and MNPs) between a silicon wafer and a glass mask separated by 10 µm thick spacers. Subsequently, the solution was polymerized using ultraviolet lamps for 1 min. A magnetic field of 10 mT was simultaneously applied to orient the reinforcing MNPs in the supporting layer. Afterwards, the spacer was removed and new 30 µm thick SU-8 spacers were fabricated on the wafer silicon wafer. By sandwiching the silicon wafer and the glass mask, a nanocomposite solution containing NIPAAm–AAm–PEDGA and MNPs was injected into the chamber. After ultraviolet polymerization (2 mins), the chamber was opened and the bilayers were released by immersion in water.

**Fabrication of compound micromachines.** The fabrication of hybrid machines with a bilayer head and monolayer flagellum requires sequential spatial patterning. First, the machine head was fabricated using the two-step photolithography process. During this first step, the flagellum part of the two-dimensional design was temporarily blocked using a plastic photo mask placed on top of the glass mask. After polymerization of the head, the flagellum of the machine was fabricated in the absence of the plastic mask.

**Data availability.** All the relevant data used to prepare this manuscript and the Supplementary Information is available upon request.

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

## Acknowledgements

This work was financially supported by the European Research Council Advanced Grant 'Microrobotics and Nanomedicine (BOTMED)' and the Swiss National Science Foundation. We are thankful to Dr Stefano Fusco, Mr Jonathan Perraudin and Dr Qi Zhang for their assistance in the development of the concept and data processing, Dr Famin Qiu, Dr Xiangzhong Chen and Dr Chengzhi Hu for fruitful discussions, Mr Naveen Shamsudhin for magnetic characterization, and the FIRST lab of ETH Zurich for technical support.

## Author contributions

H.-W.H. and M.S.S. conceived the ideas and designed the study, H.-W.H. performed the experiments, H.-W.H. and M.S.S. analysed the data, and all authors together wrote the manuscript.

**Additional information**

**Competing financial interests:** The authors declare no competing financial interests.

