## [Peer Review File · Nature Communications]

Reviewers' comments:

Reviewer #1 (Remarks to the Author):

Summary of the key results:

This paper integrates self-folding and magnetic actuation of micro-swimmers to create mobile, transformable micro robots.

Originality and interest:

This work is based off of two existing research areas: self-folding through polymer swelling, and micro-swimmer actuation through magnetic fields. The integration of the two is both novel and significant, I think primarily because it is relatively rare for self-folding to produce a shape for a functional purpose, but also because it integrates these two disparate actuators to multiply the behavior of these devices. The interplay between the two mechanisms provides the substance of this paper.

Data & methodology:

The quality of the methodology and data is sound.

Content:

1. Could you talk about why your helical robots don't reverse their direction when the magnetic rotation is reversed, but the devices in reference 6 do?
2. I'd like to see an explicit description of the methodology for the folding+locomotion steps together. Specifically, I'm wondering if these structures would eventually unfold as they cooled down, and how long that takes. Do they need to be regularly reheated during the locomotion steps? Because you talk about switching between modes, I'd be interested in seeing a video of one of these devices folding, moving, and then unfolding or refolding to another shape and moving again at a different speed, like you talk about in the 'programming transformation between propulsion strategies'.
3. On page 6 you say 'the magnetic moment of the machine is a function of volume', this makes sense but I recommend justifying it, at least with a reference.

Presentation:

4. I don't think you define NIR in the text; I assume it's near-infrared.
5. The only descriptions for the videos are in the main text. I would include captions in the supplemental, or perhaps add more explanatory text in the videos themselves. A legend would be useful describing, for instance, what those lines in the top right corner indicate (I assume the magnetic field).
6. Many of the diagrams need legends indicating what the different colored arrows are indicating.
7. Figure 2b has yellow arrows which I think indicate magnetic fields, applied at different times, but the figure is ambiguous about it. I recommend different colors, or separate subfigures to separate the alignment and locomotion steps

8. Can you define dimensionless speed for these devices? Absolute speed would also be good to include, at least for some of the examples.

9. In figure S6, the subfigures are out of order.

10. Many of the graph axes are labelled WSR; this acronym is defined in the main text, but should be spelled out or defined in the figure caption.

Typos:

Page four: 'arhitecture'

Page 5: 'Movie1'

Page 6: There is no figure 2d

Appropriate use of statistics and treatment of uncertainties:

11. The figures should mention the number of samples and what the error bars represent (I'm assuming standard deviation between measurements, but I'm not sure if it's 1 measurement of N devices, or some multiple)

Conclusions: robustness, validity, reliability

12. Along the same lines, it would be enlightening when looking at speed graphs such as 4b, if that represents all samples, or all samples that successfully folded. Was the folding process so reliable that every sample achieved a functional shape?

Suggested improvements: experiments, data for possible revision

As I mentioned above, it would be powerful to see a single video that captures both folding and locomotion, particularly if it showed locomotion in different forms so we can see how shape affects behavior.

References:

The references are thorough and well-used

Clarity and context:

There are a lot of different results presented, but they are organized nicely, and it is relatively easy to navigate. The introduction sets the paper up well.

13. I would be interested to see something in the discussion about future challenges with this work.

Reviewer #2 (Remarks to the Author):

The authors demonstrated the fabrication of bio-inspired, complicated polymer structures, which show multi-functionalities. Specifically, they employed two-layered photolithography approach to incorporate two different functions - i.e. magnetic controllability and thermo-responding characteristics - into a single compound structure.

They achieved complicated swimming motion as well as NIR induced shape change of the fabricated structure, allowing the functionality of the polymer structure much improved.

However, I do not believe that the submitted manuscript is appropriate for publication in Nature Communications. Here I attach some of my opinions:

i)

The authors already demonstrated a development of the functional structures reacting NIR stimuli, and I believe that the demonstrated work in this manuscript results from simple combination of the previous work and the use of MNPs, which have been utilized in many references.

ii)

The authors claim that the developed method employs programmable approach; however, it seems that their method has a limited controllability. Their approach such as modeling and calculation to mimic the nature-derived motile organisms seems appropriate, but it seems very difficult to be applied in somewhat general cases.

iii)

Overall, in my personal opinion, their introduction and discussion overstate what have actually achieved. Also, data organization in figures needs further improvement for better readability.

Reviewer #3 (Remarks to the Author):

Report on

Soft micromachines with programmable motility and morphology

by H.-W. Huang, M. Seman Sakar, A. J. Petruska, S. Pané and B. J. Nelson

for Nature Communication

The paper deals with new fabrication process of artificial micro-swimmers made of multi-layers of hydrogel containing magnetic nano-particles. The different layers have different elastic constants which drive buckling instabilities and allow the structure to fold in an origami manner when heated. Structures can fold and unfold depending on the heating and can move through complex oscillations in a rotating magnetic field thanks to embedded magnetic micro-beads in the structure. Another role of magnetic beads with a specific arrangement is to make the hydrogel anisotropic and influence the folding. Author call this technique "programming technique" for the folding. The folding follows the classical Timoshenko theory. A flagellum like structure and a head like structure together constitute the whole artificial swimmer.

Beside the method of fabrication itself, the main finding of the paper is that swimmers with an helical flagellum are slower than swimmers with a straight tail which oscillates in the magnetic field. I would like the author to clarify this point and tell us why it is so.

Another finding is the swimmer versatility when heated by a laser source. Authors compare this versatility with the African Trypanosome which can change its morphology depending of the biological environment. With the involved temperatures (45°C), I doubt that these artificial swimmers could work in the human body. I would prefer a speculation about future experiments like in-vitro experiments or within microfluidic devices...

I think that additional information should be added in the main text, especially concerning the nanoparticles (their nature, morphology, size, geometry of the arrangements, method to embed the beads, ...). With that respect, figure 1c is not sufficient. Since the paper is mainly centered on the method of fabrication authors, in my opinion, should give more details.

The paper is interesting because of the new fabrication method and also because of possible versatility in the swimmer morphology. It is easy to read and well written. I think the paper deserves publication in Nature Communication if above additional information could be provided.

Minor point:

NIR: acronym not defined

fig1-a: *Chlamydomonas Reinherdtii* has a rounded body and not a slender one as represented.

Page 7, authors mention a value of a dimensionless speed (375). The dimensionless speed is not defined in the main text. In fig 4b, the speed is expressed in (body length per second) and the maximum value (for the structure with a non-helical flagellum) is around 0.8 body length per sec. A clear connection between the figure and the text should be given.

REVIEWERS' COMMENTS:

Reviewer #1 (Remarks to the Author):

The authors have addressed all of my comments, and I think this paper is now ready and appropriate for publication.

Reviewer #3 (Remarks to the Author):

Authors have answered questions and comments with clarity and rigour. I do believe the paper deserves publication in Nature Comm.

Reviewer #1 (Remarks to the Author):

Summary of the key results: This paper integrates self-folding and magnetic actuation of micro-swimmers to create mobile, transformable micro robots.

Originality and interest: This work is based off of two existing research areas: self-folding through polymer swelling, and micro-swimmer actuation through magnetic fields. The integration of the two is both novel and significant, I think primarily because it is relatively rare for self-folding to produce a shape for a functional purpose, but also because it integrates these two disparate actuators to multiply the behavior of these devices. The interplay between the two mechanisms provides the substance of this paper.

Data & methodology: The quality of the methodology and data is sound.

References: The references are thorough and well-used.

Clarity and context: There are a lot of different results presented, but they are organized nicely, and it is relatively easy to navigate. The introduction sets the paper up well.

We are glad the reviewer found the approach and results interesting and hope that the following explanations and additional experiments provide adequate clarification.

1. Could you talk about why your helical robots don't reverse their direction when the magnetic rotation is reversed, but the devices in reference 6 do?

The reviewer raises an important and astute question that we had not carefully dissected. Corkscrew motion, as demonstrated in reference 6 (Zhang, L. *et al.* Artificial bacterial flagella: Fabrication and magnetic control. *Appl. Phys. Lett.* 94, 064107, 2009), is completely reversible. When the direction of rotation of the magnetic field is reversed, the robot instantaneously starts to move in the opposite direction (as inertial forces do not play a role at low Reynolds number). The micromachines shown in reference 6 are rigid, and they do not deform during locomotion. However, the results we have shown in this manuscript were generated with micromachines with flexible bodies, which led to bending during locomotion. Due to this bending, the orientation of the rotational axis of the rotating body changes, which is known as precession. While the machine is moving forward, the body is tilted with respect to the direction of motion, and it precesses tracing out a helical trajectory. In the reverse mode, the precession still points in the

forward direction. Due to this attenuation of motion in the reverse mode, the machine starts to move backwards only when the corkscrew motion is stronger than the precession, which only happens at increasing frequencies. The direction of enhancement/attenuation depends on the chirality of the helical tail. The same mechanism was reported in the helical motion of the *Caulobacter crescentus* and the source of the bacterial cell body precession is believed to be associated with the flexibility of the hook that connects the flagellum to the cell body (Liu, B. *et al.* Helical motion of the cell body enhances *Caulobacter crescentus* motility. Proc. Natl. Acad. Sci. USA 111, 11252-11256, 2014).

To respond to Reviewer 1's question and to systematically demonstrate the contribution of precession, we fabricated three different micromachines: a non-deformable (relatively stiff) machine with only a helical body, a non-deformable machine with helical tail and tubular head, and a deformable (relatively soft) machine with helical tail and tubular head (see below). The stiffness of the machines was tuned by playing with the cross-linking degree of the hydrogel. The machines were driven at 2 Hz and 3 Hz in clockwise (CW) and counterclockwise (CCW) direction. The first and second type of micromachines instantaneously changed direction of motion when the direction of magnetic field rotation was reversed even at very low frequencies. The existence of a tubular head generates higher drag force, which reduces the motility in both directions. Therefore, micromachines with tubular heads swim with lower velocity. The motility of these soft micromachines is comparable to rigid micromachines (i.e. the machines used in reference 6).

The motility of the third type of soft micromachine is composed of the precession of the body and the propulsion generated by the tail. These machines follow a helical trajectory, and precession enhances the forward motility (CCW rotation) while attenuating the backward motility (CW rotation) as expected. At low rotating frequencies, the motility of the machine is dominated by the precession of the body because of the large precession angle and low propulsion force generated by the tail. As a result, the machine moves forward regardless of the direction of magnetic field rotation (both CW and CCW). Increasing the frequency reduces the precession angle and increases the propulsive force of the tail. Therefore, the motility is dominated by corkscrew motion at high frequencies and this motion is reversible (machines reverse their direction when the direction of rotation is reversed).

The data shown below has now been included in Supplementary Figure S7, and we also added a new supplementary movie (Supplementary Movie 5) to visualize the motility of these three different types of micromachines.

2. I'd like to see an explicit description of the methodology for the folding+locomotion steps together. Specifically, I'm wondering if these structures would eventually unfold as they cooled down, and how long that takes. Do they need to be regularly reheated during the locomotion steps? Because you talk about switching between modes, I'd be interested in seeing a video of one of these devices folding, moving, and then unfolding

or refolding to another shape and moving again at a different speed, like you talk about in the 'programming transformation between propulsion strategies'.

Following the reviewer's suggestion, we created a new movie that visualizes all the steps from initial folding process to magnetically driven locomotion (see Supplementary Movie 6). The movie first shows the folding of swarms of microfabricated hydrogel layers due to differential swelling. Next, we zoom in and follow the undulatory motion of one of these micromachines driven by rotating magnetic fields. The same machine then goes through a shape transformation triggered by an increase in ambient temperature, and the tail wraps around the head as a result of the unfolding and subsequent refolding processes. The machine not only loses its propulsive force generated by the tail during this transformation, the magnetization of the tubular head also changes. As a result, the machine starts to demonstrate a completely different locomotion mode, namely tumbling.

The shape transformation is reversible and the micromachines will eventually unfold when they are cooled down. If the ambient temperature is kept at the same higher level, they will of course retain their shape and motility. This characteristic can be considered as an adaptation to changes in the environmental conditions. Alternatively, the machines must be continuously exposed to the NIR stimuli to sustain the transient morphological state. We measured the time needed to complete folding, refolding and unfolding steps (see below). First, micromachines were transferred from a solution at 22°C to a solution at 45°C, and their unfolding and refolding behavior was monitored. The micromachines were then transferred back to the solution at 22°C to record their transformation to their original shape. In a second set of experiments, micromachines were exposed to near-infrared light (2 mW and 808 nm) in a solution at 22°C until the completion of unfolding and refolding processes. The exposure was then turned off and machines transformed back to their original shape while they were cooling down. The measurements were repeated three times from three different micromachines with identical shapes and the average values were calculated. The time-lapse images shown below are the screen shots from the movies recorded during these experiments. While, the shape transformation triggered by NIR at the current settings takes longer than directly submerging machines into a warmer solution, laser heating can be more effective at higher laser power or MNP concentrations. Interestingly, unfolding events happening in both directions (from 22°C to 40°C and from 45°C to 40°C) are significantly faster than the folding events (from 40°C to 45°C and from 40°C to 22°C). This data is now displayed in Supplementary Figure S5.

3. On page 6 you say 'the magnetic moment of the machine is a function of volume', this makes sense but I recommend justifying it, at least with a reference.

We reworded that sentence for better clarity as follows: “The magnetic moment of the machine is a function of the MNP volume, and the resulting motion is dominated by the component with the most MNPs”. We cited a textbook on magnetism to justify this statement (Reference 27: Spaldin, N. A. Magnetic Materials Fundamentals and Applications. 2nd edn, Cambridge University Press, 2010). In addition, we measured the magnetization of nanocomposites with varying nanoparticle concentration (see below) and experimentally verified that magnetization of our microstructures depends on the volume of incorporated MNPs. This data is now displayed in Supplementary Figure S5.

4. I don't think you define NIR in the text; I assume it's near-infrared.

That is correct. We apologize for forgetting to mention this important information. We defined the acronym where it was first used in the introduction.

5. The only descriptions for the videos are in the main text. I would include captions in the supplemental, or perhaps add more explanatory text in the videos themselves. A legend would be useful describing, for instance, what those lines in the top right corner indicate (I assume the magnetic field).

We included the captions in the supplementary information and directly added more explanatory text in all of the movies. That is correct. The red line indicates the direction of applied magnetic field and the blue line indicates the swimming direction. We clearly described these lines in the movies.

6. Many of the diagrams need legends indicating what the different colored arrows are indicating. Figure 2b has yellow arrows, which I think indicate magnetic fields, applied at different times, but the figure is ambiguous about it. I recommend different colors, or separate subfigures to separate the alignment and locomotion steps

We now provide a well-defined color code and terminology for all the arrows, lines and labels indicating alignment of nanoparticles, the orientation of the applied magnetic field during programming, the folding and re-folding axis, and the orientation of the applied magnetic field during locomotion. In Figure 2 (see below), “folding axis 1” now indicates the folding direction of the head and “folding axis 2” indicates the folding direction of the tail. The folding axis of the head of the micromachine shown in Figure 2b is determined by the particle alignment and is not affected by the position of the tail.

7. Can you define dimensionless speed for these devices? Absolute speed would also be good to include, at least for some of the examples.

The dimensionless speed is defined as $U_{Dl} = U/fL \times 10^3$ where U is the forward velocity, f is the rotating frequency and L the length of machine head (Pak, O. S., Gao, W., Wang, J. & Lauga, E. High-speed propulsion of flexible nanowire motors: Theory and experiments. *Soft matter* 7, 8169-8181, 2011). This definition is now incorporated into the maintext. We decided to show the absolute speed in Figure 4b and provided the dimensionless speed in Supplementary Figure S6.

8. In figure S6, the subfigures are out of order.

We apologize for the mistake. The order is now corrected.

9. Many of the graph axes are labelled WSR; this acronym is defined in the main text, but should be spelled out or defined in the figure caption.

WSR (weight to swelling ratio) is now defined in all the relevant figure captions.

10. Typos: Page four: 'arhitecture', Page 5: 'Movie1', Page 6: There is no figure 2d

The typographical errors are now corrected.

11. The figures should mention the number of samples and what the error bars represent (I'm assuming standard deviation between measurements, but I'm not sure if it's 1 measurement of N devices, or some multiple)

The error bars in all the figures represent standard deviations. The standard deviations are calculated from six different measurements from each of the three different micromachines with identical shapes (a total of 18 measurements). We added this information to the maintext.

12. Along the same lines, it would be enlightening when looking at speed graphs such as 4b, if that represents all samples, or all samples that successfully folded. Was the folding process so reliable that every sample achieved a functional shape?

The throughput of the folding process is close to 100% and they are all functional. However, the shapes of the micromachines slightly differ due to small changes in material composition, layer thickness, UV exposure, and the alignment of magnetic nanoparticles. The swimming performance of the micromachines is sensitive to the shape and stiffness of the tail. Therefore, we worked with micromachines with almost the same shape when we tested the role of morphology in motility.

13. As I mentioned above, it would be powerful to see a single video that captures both folding and locomotion, particularly if it showed locomotion in different forms so we can see how shape affects behavior.

Following the reviewer's suggestion, we created a new movie that visualizes all the steps from initial folding process to magnetically driven locomotion (Supplementary Movie 6). In this new video we explicitly show that the soft micromachine transforms from a long slender form that is radially magnetized to a short stumpy form that is magnetized along its long axis. As a result, its locomotion changes from corkscrew motion to tumbling motion.

14. I would be interested to see something in the discussion about future challenges with this work.

Following the suggestion of the reviewer, we expanded our discussion with an analysis of future challenges and opportunities (see below).

In summary, we present a practical method for building functional micromachines with complex geometries through programmable self-assembly. Controlling the morphology of the folded mechanism by patterning multiple hydrogel layers limits the scalability of the approach as this technique requires patterning of high aspect ratio structures. Generation of stiffness gradients within a hydrogel monolayer with gravitational sedimentation allow us to initiate self-folding, and more sophisticated devices at the microscale can be fabricated by magnetically controlling the distribution of MNPs along the thickness direction. Although we focus on bioinspired corkscrew motion, our approach is not limited to propulsion generated by a rotating passive flagellum. More complex swimming strategies can be realized by engineering parts with a different magnetic easy axis (Kim, J. *et al.* Programming magnetic anisotropy in polymeric microactuators. *Nat. Mater.* 10, 747-752, 2011) and application of spatiotemporally varying magnetic fields. While thermally triggered deformation is completely reversible for the presented prototypes, micromachines with multiple stable conformations can be engineered by combining local heating with engagement initiated by internal magnetic forces among body parts (Hawkes, E. *et al.* Programmable matter by folding. *Proc. Natl. Acad. Sci. USA*, 107, 12441-12445, 2010).

Reviewer #2 (Remarks to the Author):

Summary of the key results: The authors demonstrated the fabrication of bio-inspired, complicated polymer structures, which show multi-functionalities. Specifically, they employed two-layered photolithography approach to incorporate two different functions - i.e. magnetic controllability and thermo-responding characteristics - into a single compound structure. They achieved complicated swimming motion as well as NIR induced shape change of the fabricated structure, allowing the functionality of the polymer structure much improved. However, I do not believe that the submitted manuscript is appropriate for publication in Nature Communications.

1. The authors already demonstrated a development of the functional structures reacting NIR stimuli, and I believe that the demonstrated work in this manuscript results from simple combination of the previous work and the use of MNPs, which have been utilized in many references.

As pointed out by reviewer 1 and 3, who both found the work to be “novel and significant”, it is relatively rare for self-folding to produce a shape that is functional. In addition to autonomously folding complex micromachines with multiple body parts, alignment of magnetic nanoparticles allows us to selectively tune magnetic anisotropy and structural properties of each compartment. These emergent features make the work an original contribution, which does not build from a simple combination of previous accomplishments; they open up a new design space for building micromachines with programmable architecture and controllable locomotion. In our previous work, we utilized NIR stimuli for heating up thermoresponsive hydrogel layers doped with graphene oxide to induce unfolding of simple structures. In the present, with spatially controlled distribution and alignment of MNPs, we demonstrate a true transformation of machines from one functional state to a state with a completely different morphology and functionality. We also succeed to initiate local transformation with selective exposure of body parts.

2. The authors claim that the developed method employs programmable approach; however, it seems that their method has a limited controllability. Their approach such as modeling and calculation to mimic the nature-derived motile organisms seems appropriate, but it seems very difficult to be applied in somewhat general cases.

It is true that no detailed design methodology currently exists to mimic nature-derived motile organisms, but this work represents a step in the direction and points to paths forward using detailed models for eventually creating such a design methodology. As described in the

maintext, in addition of alignment of MNPs, generating gradients of MNPs via magnetic field gradients can add a whole new level of programming capabilities especially at smaller scale.

3. Overall, in my personal opinion, their introduction and discussion overstate what have actually achieved. Also, data organization in figures needs further improvement for better readability.

We have made several changes in the paper based on specific comments from the other reviewers. We believe the organization and clarity of presentation is significantly improved after these changes.

Reviewer #3 (Remarks to the Author):

Summary of the key results: The paper deals with new fabrication process of artificial micro-swimmers made of multi-layers of hydrogel containing magnetic nano-particles. The different layers have different elastic constants, which drive buckling instabilities and allow the structure to fold in an origami manner when heated. Structures can fold and unfold depending on the heating and can move through complex oscillations in a rotating magnetic field thanks to embedded magnetic micro-beads in the structure. Another role of magnetic beads with a specific arrangement is to make the hydrogel anisotropic and influence the folding. Author call this technique "programming technique" for the folding. The folding follows the classical Timoshenko theory. A flagellum like structure and a head like structure together constitute the whole artificial swimmer.

Originality and interest: The paper is interesting because of the new fabrication method and also because of possible versatility in the swimmer morphology.

Clarity and context: It is easy to read and well written.

I think the paper deserves publication in Nature Communication if below additional information could be provided.

We thank the reviewer for his/her enthusiastic support for publication of this work and have strived to incorporate his/her comments in the revised manuscript.

1. Beside the method of fabrication itself, the main finding of the paper is that swimmers with a helical flagellum are slower than swimmers with a straight tail which oscillates in the magnetic field. I would like the author to clarify this point and tell us why it is so.

Our finding is in line with published results. Previous work has experimentally demonstrated that flexible structures (fabricated as hinged structures or microfilaments) driven by oscillating or rotating magnetic fields swim significantly faster than rigid helical propellers performing corkscrew motion. Theoretical analysis and experiments with flagellated bacteria have shown that the propulsion of a helix in a viscous fluid produced by its rotation is very inefficient, both kinematically (small forward motion per rotation) and energetically (only 2% percent of the total work done by the helix is transmitted to propulsive work). On the other hand, experiments with unflagellated bacterium *Caulobacter crescentus* has shown that due to the flexibility of the hook the cell body is tilted with respect to the direction of motion, and it precesses, tracing out a helical trajectory. The helical motion of the cell body generates thrust and enhances forward swimming motility, which is proportional to the precession angle. We did not engineer a special hook that connects the tail to the head. For the chosen material composition and design parameters, micromachines with helical tails exhibit high symmetry in rotating axis, which stabilizes the precession of head and results in a relatively straight trajectory. However, micromachines with planar tails swim with a significantly higher precession angle and follow a 3D helical trajectory as shown in Figure 4 and Supplementary Figure S6. Therefore, the superior swimming performance of micromachines with planar tails is coming from the synergetic propulsion generated by the precession of the head and rotation of the tail. On the other hand, the backward velocity is inversely proportional to the precession angle due to the unidirectional propulsion provided by precession. For a detailed description of this swimming behavior, please see the answer to first reviewers' first question.

2. Another finding is the swimmer versatility when heated by a laser source. Authors compare this versatility with the African Trypanosome which can change its morphology depending of the biological environment. With the involved temperatures (45 °C), I doubt that these artificial swimmers could work in the human body. I would prefer a speculation about future experiments like in-vitro experiments or within microfluidic devices...

Previous work has shown that the transition temperature can be adjusted by changing the type (hydrophobic vs hydrophilic) and concentration of co-polymers incorporated into the nanocomposite. For example, as shown in the supplementary Figure S10a (also shown below), by removing the hydrophilic co-polymer (Acrylamide, AAm) from the nanocomposite solution, we

can decrease the transition temperature from 45 °C to 32 °C. A further reduction in transition temperature is achievable (LCST=9.8°C) by incorporating a hydrophobic co-polymer (N-tertbutylacrylamide, NtBAAm) into the solution with the ratio of 50:50 between NIPAAm and NtBAAm (Klouda, L. & Mikos, A. G. Thermoresponsive hydrogels in biomedical applications - a review. *Eur J Pharm Biopharm.* 68, 34-45, 2008; Doorty, K. B. *et al.* Poly(N-isopropylacrylamide) co-polymer films as potential vehicles for delivery of an antimetabolic agent to vascular smooth muscle cells. *Cardiovascular Pathology* 12, 105-110, 2003). We incorporated this information along with the accompanying citations into the manuscript.

3. I think that additional information should be added in the main text, especially concerning the nano-particles (their nature, morphology, size, geometry of the arrangements, method to embed the beads, ...). With that respect, figure 1c is not sufficient. Since the paper is mainly centered on the method of fabrication authors, in my opinion, should give more details.

MNPs are commercially available, and we used them without further modification. The purity of the iron oxide nanoparticles (Fe_3O_4) is up to 98%, and they are coated with 1 wt% polyvinylpyrrolidone (PVP) to prevent agglomeration in the pre-gel solution. Transmission electron microscopy (TEM) imaging shows that the average size of the nanoparticles is around

30 nm (see below). We performed vibrating sample magnetometer (VSM) measurements with a scanning magnetic range between -30 mT and 30 mT, which covers the magnetic fields used in all of our experiments. The VSM results show that the largest hysteresis loop and magnetization are in the direction parallel to the alignment of MNPs and the smallest ones are in the direction perpendicular to the alignment. The hysteresis loop appears between the parallel and perpendicular alignment for the random distribution. Therefore, it is safe to claim that the alignment of MNPs determines the magnetic anisotropy of the nanocomposite. In addition, we performed low temperature scanning electron microscopy (Cryo-SEM), and the images show that nanoparticles are uniformly distributed throughout the hydrogel polymer network. Finally, we also provide the optical micrographs of the hydrogel nanocomposites with and without magnetic alignment. Images clearly show that the MNPs are aligned within the helical micromachine along the programmed axis, in both parallel and perpendicular directions. These descriptions are now incorporated into the maintext and the data is presented in a new figure, Supplementary Fig. S9.

We have also significantly revised Figure 1 to provide a detailed graphical description of the fabrication process and following steps on self-folding, magnetic actuation and thermally-induced shape transformation (see below).

4. NIR: acronym not defined

We apologize for forgetting to mention this important information. NIR stands for near-infrared. We defined the acronym where it was first used in the introduction.

5. fig1-a: Chlamydomonas Reinherdtii has a rounded body and not a slender one as represented.

We corrected this mistake and redrew the schematic of *Chlamydomonas Reinherdtii*.

6. Page 7, authors mention a value of a dimensionless speed (375). The dimensionless speed is not defined in the main text. In fig 4b, the speed is expressed in (body length per second) and the maximum value (for the structure with a non-helical flagellum) is around 0.8 body length per sec. A clear connection between the figure and the text should be given.

All the values are now clearly defined in the text including the dimensionless speed, and connections are established between the text and the figures. The dimensionless speed is defined as $U_{DI} = U/fL \times 10^3$ where U is the forward velocity, f is the rotating frequency and L the length of machine head. This definition is now incorporated into the maintext. We decided to show the absolute speed in Figure 4b and provided the dimensionless speed in Supplementary Figure S6.